# The paradoxical effect of oral prednisone in granulomatous lobular mastitis: An observational cohort study

**Haiyan Zhang[1], Ruiyang Wu[2], Jin Chen[2], Jinyan Feng[1], Jing Luo[1]***

**1** Breast Centre, West China Second University Hospital, Chengdu, China, **2** Department of Breast and Thyroid, Sichuan Provincial Hospital for Women and Children (Affiliated Women and Children's Hospital of Chengdu Medical College), Chengdu, China

\* 1318497642@qq.com

## Abstract

### Background

Despite being a cornerstone therapy for granulomatous lobular mastitis (GLM), the use of oral prednisone lacks a standardized protocol due to undefined optimal parameters and insufficient large-scale evidence regarding its impact on disease recurrence.

### Objectives

The retrospective cohort study comprehensively assessed the efficacy of oral prednisone for GLM by utilizing a large-scale cohort. It aimed to identify the optimal therapeutic protocol, with the ultimate aim of establishing a robust evidence base to inform standardized clinical guidelines.

### Methods

A cohort of 614 patients diagnosed with GLM was included in this study from January 2017 to December 2024. A three-dimensional evaluation of treatment timing (symptom-to-treatment interval), duration, and initial dosage (both absolute and weight-adjusted) was performed to identify prognostic factors and optimal cutoffs.

### Results

Oral prednisone was identified as an independent risk factor for recurrence (HR = 2.534, 95% CI: 1.615–3.975, P < 0.001). Survival curve analysis revealed that the 1-year recurrence rate was significantly higher in the oral prednisone group than in the non-prednisone group (29.4% vs. 14.6%, P = 0.001). Subgroup analysis showed that the long-interval, low-dose group (symptom-to-treatment interval > 6 weeks, dosage ≤ 0.4 mg/kg/day) yielded the most favorable results. The one-year recurrence rate of this group (6.2%) was significantly lower than other groups (all

**Data availability statement:** The datasets generated and analyzed in this study are not publicly available as they belong to a proprietary clinical database. Access can be granted for research purposes upon formal request. Interested researchers should contact the institutional data steward, Director Hu (Department of Breast and Thyroid Surgery), at 765994388@qq.com. Applicants were asked to specify the intended use of the data and to commit to compliance with the terms of use. Any data sharing was contingent upon final approval from the relevant institutions and the signing of a legally binding data use agreement.

**Funding:** The author(s) received no specific funding for this work.

**Competing interests:** The authors have declared that no competing interests exist.

$P<0.05$) and comparable to the non-prednisone group (14.6%, $P=0.233$). Conversely, the short-interval group (symptom-to-treatment interval: 0–6 weeks) demonstrated the highest one-year relapse rate (33.5%), significantly exceeding that of the non-prednisone group ($P<0.001$).

## Conclusions

While oral prednisone therapy may be associated with an increased relapse risk in GLM, an optimized protocol—initiating treatment after a 6-week symptom interval at a dose of ≤ 0.4 mg/kg/day—was identified to mitigate this risk. These findings highlight the paramount importance of treatment timing and dosage, offering a data-driven strategy to refine current therapeutic practices. Prospective studies are warranted to validate this proposed regimen.

---

## Introduction

Granulomatous lobular mastitis (GLM) is a rare, benign, and chronic inflammatory disease of the breast with an unknown etiology [1,2]. Nonetheless, emerging evidence from large-scale, population-based studies indicates a rising diagnostic frequency of GLM in recent years, suggesting it is transitioning from a rare entity to a more commonly encountered clinical condition [3]. GLM typically presents with rapid-onset inflammation that progresses to abscesses and chronic, disfiguring sinus tracts, which often results in a severe and protracted disease course persisting for one to two years [4,5].

There is no universally agreed-upon therapeutic strategy for GLM, and its management is actively debated. Nonetheless, corticosteroid-based regimens have established themselves as a cornerstone of treatment due to their profound anti-inflammatory and immunomodulatory effects [6–9]. Corticosteroid therapy for GLM encompasses both systemic and topical approaches [10]. Although local steroid therapy (e.g., intralesional injection) has been reported as a useful treatment modality [6,11] and offers the potential advantages of direct drug delivery and reduced systemic exposure compared to oral therapy, its widespread clinical adoption remains limited by the need for professional administration, injection-related discomfort, and a lack of large-scale comparative evidence. Systemic steroid therapy, most commonly employing prednisone, is more frequently utilized [7]. However, this approach faces several major challenges. First, recurrence rates following systemic corticosteroid treatment are notably high, with reported figures ranging widely from approximately 25% to over 45% across different studies [4,7,12,13]. Second, a proportion of patients experience adverse effects—such as weight gain, hirsutism, hyperglycemia, and iatrogenic cushing—which frequently impair treatment compliance and limited therapy duration [14]. Finally, owing to the limited evidence from small case series [1,4,7,12–17], key therapeutic parameters—including the optimal starting dose, treatment timing, and total treatment duration—have not been established, leaving clinicians without a standardized, evidence-based regimen.

Given the lack of consensus and the highly variable practices in systemic corticosteroid therapy, this study comprehensively assessed the efficacy of oral prednisone for GLM by utilizing a large-scale cohort. The aim was to identify the optimal therapeutic protocol, with the ultimate goal of establishing a robust evidence base to inform standardized clinical guidelines.

## Materials and methods

### Data and samples

We conducted a large-scale cohort study using data from a proprietary non-puerperal mastitis database established in March 2022 at a major tertiary maternity hospital. The database enrolled patients who were diagnosed with non-puerperal mastitis (NPM) according to the Chinese expert consensus [18] who have been treated at the institution since January 2017. Prospective data collection commenced in March 2022, while data prior to this date were collected retrospectively. Data collection followed a standardized protocol throughout both phases. Prospective data were recorded in real time using predefined forms, while retrospective data were systematically re-extracted from original records by two independent investigators applying the same variable definitions. This process was guided by an internal database manual, with institutional details anonymized for review.

The diagnosis of GLM for all included cases was established through a comprehensive assessment, integrating clinical presentation and imaging findings, and was definitively confirmed by histopathological examination. Specifically, GLM cases were identified and confirmed based on pathological reports, after the exclusion of infectious etiologies (e.g., tuberculosis) and other specific granulomatous diseases, consistent with expert diagnostic pathways [19]. Subsequently, to focus the analysis on the efficacy of oral prednisone, the following three study-specific exclusion criteria were applied to the initial GLM cohort: (1) patients diagnosed with plasma cell mastitis (a distinct NPM subtype); (2) patients with missing body weight records (which precluded calculation of weight-adjusted doses); and (3) patients who received corticosteroid therapy other than oral prednisone.

From an extraction on September 10, 2025, we identified 710 patients diagnosed between January 2017 and December 2024. After applying exclusion criteria, we sequentially excluded 44 patients with plasma cell mastitis, 8 with unknown body weight, and 44 who received non-prednisone corticosteroid therapy, resulting in a final analytical cohort of 614 patients with GLM. The protocol for this cohort study, which utilized a private database, was approved by the Institutional Ethics Committee in March 2022 (Approval No. 20220330−024) prior to its initiation. The study was conducted in compliance with the ethical principles of the Declaration of Helsinki (2013 revision). As the research involved only the analysis of pre-existing, anonymized data without any patient intervention, the ethics committee granted a waiver of informed consent.

This study defined recurrence as the appearance of NPM symptoms (such as redness, swelling, pain, or a palpable mass) in the ipsilateral or contralateral breast following documented initial clinical improvement or cure. Cure was defined as complete resolution of symptoms, disappearance of palpable masses, and negative findings on both physical examination and imaging studies [18]. All potential recurrence events during follow-up were identified and adjudicated by the study's database manager, a senior breast disease specialist, applying the predefined criteria above. The time-to-event analysis measured the duration from diagnosis to recurrence or the final follow-up, with all durations rounded to the nearest month. The final follow-up date was September 1, 2025.

The laboratory parameters analyzed in this study (white blood cell count, C-reactive protein, prolactin) were obtained from multiple tests conducted throughout the patients' clinical course. Given that not all patients underwent testing at initial presentation, and to establish a consistent measure of disease activity, the following approach was adopted: the highest recorded value from all available test results for each patient, spanning from their first visit for the index episode to the last follow-up, was extracted and analyzed. This method is based on the clinical rationale that the peak level of inflammation or hormonal activity during the disease course may better reflect its maximum biological intensity and was

used for subsequent prognostic analysis. Hyperlipidemia was defined as meeting at least one of the following: total cholesterol ≥ 220 mg/dL, low-density lipoprotein cholesterol ≥ 140 mg/dL, high-density lipoprotein cholesterol < 40 mg/dL, or triglycerides ≥ 150 mg/dL [20].

This study employed two main treatment modalities: pharmacological therapy and surgical intervention. The pharmacological strategies encompassed a range of antimicrobials (fluoroquinolones, penicillins, cephalosporins, macrolides, nitroimidazoles), anti-tuberculosis agents (isoniazid, rifampin, or ethambutol), oral prednisone (the regimen, including the starting dose and tapering schedule, was determined by the attending physician based on clinical judgment, commonly involving an initial dose followed by a taper of 5 mg every 3–7 days until discontinuation), and bromocriptine. Surgical interventions involved procedures for abscess drainage (e.g., aspiration, vacuum-assisted drainage, open debridement) and surgical excision (via vacuum-assisted biopsy or open surgery).

## Statistical analysis

Continuous variables were expressed as mean and standard deviation (SD), while categorical variables were presented as frequencies and percentages. To investigate the impact of oral prednisone treatment on relapse in patients with GLM, two analytical methods were employed. First, a Cox proportional hazards regression model was used to identify potential predictors of disease relapse, with a focus on oral prednisone therapy. Second, patients were divided into two groups: the oral prednisone group (patients who received oral prednisone at the entire course of the disease) and non-prednisone group (patients who never received any corticosteroid therapy at the entire course of the disease, including oral prednisone or local treatment). Propensity score matching (PSM) was used to balance covariates that differed between the two groups. The survival curves of the two groups before and after matching were compared using the Breslow test to evaluate the treatment effect.

We further analyzed the prognostic role of oral prednisone through three dimensions: (1) treatment timing, by stratifying patients into short-interval and long-interval groups based on the time from symptom onset to treatment initiation; (2) treatment duration, by categorizing patients into short-duration and long-duration groups according to the total course of therapy (excluding maintenance doses); and (3) treatment dosage (excluding maintenance doses). To comprehensively evaluate the dose-effect relationship, the dosage was analyzed using two complementary metrics: the absolute initial dose (mg/day) and the weight-adjusted initial dose (mg/kg/day). The absolute dose reflects the overall intensity of drug exposure, whereas the weight-adjusted dose accounts for inter-patient physiological differences, providing a more personalized and clinically relevant pharmacologic measure. PSM was performed using the MatchIt package in R [21], implementing a 1:1 nearest-neighbor algorithm with a caliper width of 0.1. The optimal cutoff values for the above groups were identified using X-tile software (version 3.6.1, Yale University School of Medicine, New Haven, CT, USA) [22], which applies the minimum P-value method from survival analysis. This X-tile analysis was considered hypothesis-generating; all suggested thresholds were independently validated via time-trend and multi-dimensional subgroup analyses. Based on this risk stratification, subgroup analyses were conducted to identify the optimal oral prednisone strategy (detailed in Fig 1). All hypothesis tests were two-sided, with a P-value of less than 0.05 considered statistically significant. The statistical analyses were performed using R software (version 4.2.2; R Foundation for Statistical Computing, Vienna, Austria), and the survival curves were plotted using Stata (version 19; StataCorp LLC, College Station, TX, USA).

## Results

### Descriptive characteristics and prognostic factor analysis

This study enrolled a total of 614 GLM patients with a mean age of 31.4 ± 4.8 years, and all participants were female. The mean follow-up duration was 13.9 ± 14.9 months. The one-year recurrence rate was 23.9%. Univariable and multivariable COX regression analyses (Table 1) identified the following factors as independent predictors of recurrence: affected side (P < 0.001), white blood cell count (P < 0.001), antitubercular therapy (HR = 0.394, 95% CI: 0.264–0.588, P < 0.001),

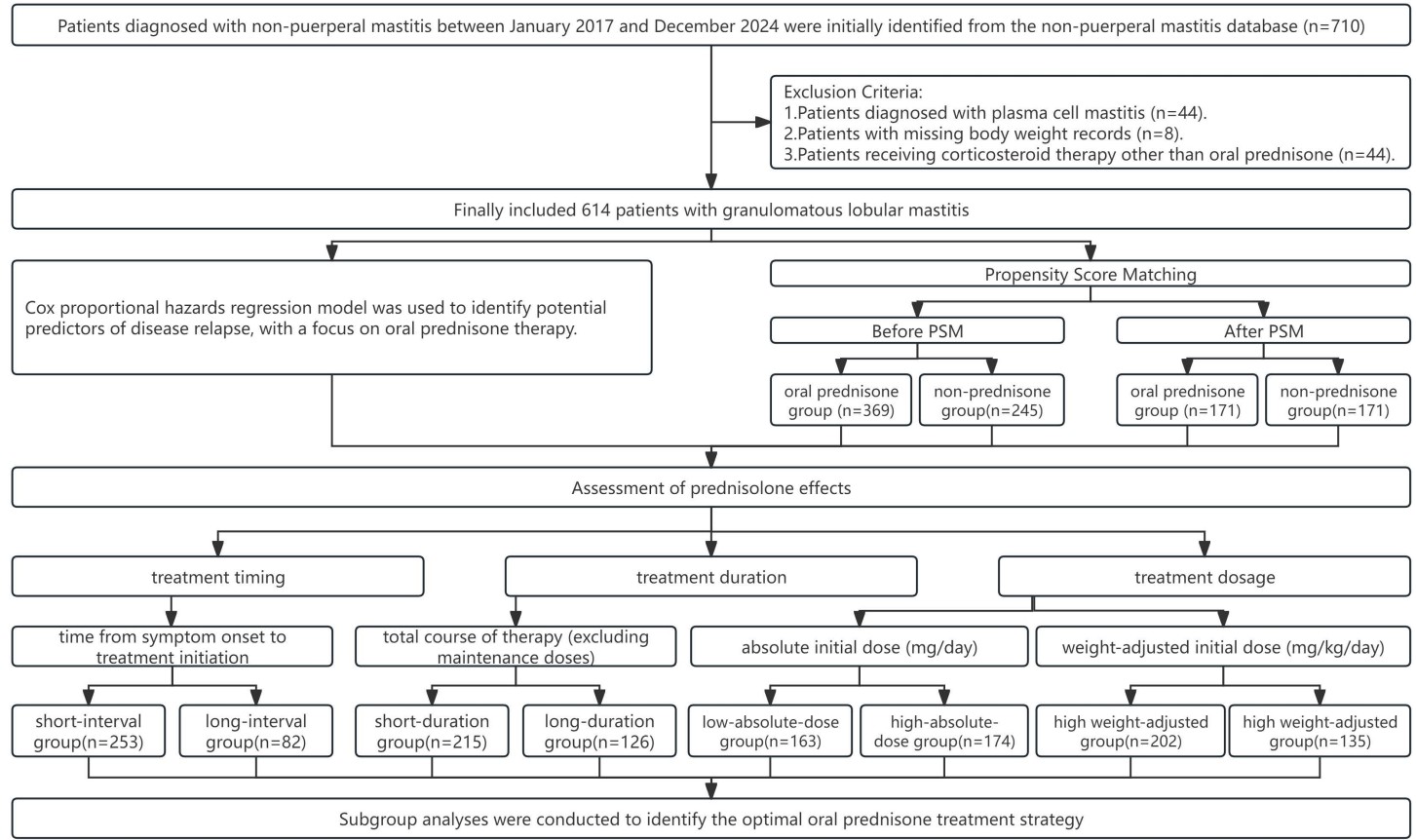

**Fig 1. Research workflow of this study.**

oral prednisone (HR = 2.534, 95% CI: 1.615–3.975, P < 0.001), and abscess drainage (HR = 0.506, 95% CI: 0.300–0.854, P = 0.011).

To evaluate the robustness of our findings with respect to the data collection mode, we conducted a sensitivity analysis using only the prospectively collected data from March 2022 onward (n = 374). This approach aimed to exclude potential influences arising from variations in completeness or consistency inherent in the earlier retrospective records. Univariable and multivariable Cox regression analyses performed on this prospective cohort (see **S1 Table**) yielded results highly consistent with the primary analysis. Affected side (P < 0.001), white blood cell count (P = 0.001), antitubercular therapy (HR = 0.433, 95% CI: 0.225–0.833, P = 0.012), oral prednisone (HR = 2.645, 95% CI: 1.332–5.253, P = 0.005), and abscess drainage (HR = 0.439, 95% CI: 0.229–0.839, P = 0.013) remained independent predictors of recurrence. The direction of effect (protective or risk) and the magnitude of the hazard ratios for these predictors were closely aligned with those obtained from the primary analysis of the entire cohort (n = 614).

## Oral prednisone

After stratifying the 614 patients into an oral prednisone group (n = 369) and a non-prednisone group (n = 245) based on treatment exposure, survival analysis before PSM revealed a significantly higher 1-year recurrence rate in the oral predni-sone group (29.4% vs. 14.6%, P = 0.001, **Fig 2A**). To address potential confounding, 1:1 PSM was performed, balancing all baseline characteristics (e.g., lesion count, abscess presence, white blood cell count, and prior treatments) between

**Table 1. Univariable and multivariate analysis of prognostic factors for 614 patients with granulomatous lobular mastitis.**

| Factors | N(%)/x±s | Univariate | Multivariate | |
|---|---|---|---|---|
| | | P value | HR(95%CI) | P value |
| Age at diagnosis (year) | 31.4±4.8 | 0.126 | 0.940(0.905-0.977) | 0.002 |
| Weight (kg) | 60.5±10.3 | 0.292 | 1.012(0.995-1.029) | 0.171 |
| Affected side | | <0.001 | | <0.001 |
| left | 319(52.0) | | Ref | |
| right | 284(46.3) | | 1.987(1.402-2.816) | <0.001 |
| bilateral | 11(1.8) | | 11.275(4.479-28.380) | <0.001 |
| Days to first visit (days) | 19.1±28.5 | 0.093 | 0.985(0.977-0.993) | <0.001 |
| Maximum lesion diameter on ultrasound | | 0.059 | | 0.021 |
| < 4 cm | 239(38.9) | | Ref | |
| ≥ 4 cm | 337(54.9) | | 1.611(1.081-2.400) | 0.019 |
| unknown | 38(6.2) | | 2.407(1.029-5.629) | 0.043 |
| Ultrasound-detected lesion count | | 0.806 | | 0.190 |
| solitary | 97(15.8) | | Ref | |
| multiple | 478(77.9) | | 0.928(0.558-1.542) | 0.772 |
| unknown | 39(6.4) | | 0.415(0.159-1.078) | 0.071 |
| Mammary abscess | | 0.271 | | 0.710 |
| no | 101(16.4) | | Ref | |
| yes | 513(83.6) | | 1.120(0.618-2.030) | |
| Microabscess | | 0.087 | | 0.094 |
| no | 234(38.1) | | Ref | |
| yes | 243(39.6) | | 0.653(0.436-0.978) | 0.039 |
| unknown | 137(22.3) | | 0.978(0.583-1.641) | 0.933 |
| White blood cell | | <0.001 | | <0.001 |
| < 10*10^9/L | 336(54.7) | | Ref | |
| ≥ 10*10^9/L | 259(42.2) | | 1.784(1.205-2.641) | 0.004 |
| unknown | 19(3.1) | | 5.516(2.508-12.134) | <0.001 |
| C-reactive protein | | 0.096 | | 0.021 |
| < 10 mg/L | 221(34.4) | | Ref | |
| ≥ 10 mg/L | 155(25.2) | | 1.023(0.623-1.681) | 0.928 |
| unknown | 248(40.4) | | 1.755(1.121-2.748) | 0.014 |
| Prolactin | | <0.001 | | 0.070 |
| ≤ 650 uIU/ml | 391(63.7) | | Ref | |
| > 650 uIU/ml | 156(25.4) | | 0.922(0.575-1.478) | 0.735 |
| unknown | 67(10.9) | | 1.909(1.072-3.398) | 0.028 |
| Hyperlipidemia | | 0.590 | | 0.905 |
| no | 194(31.6) | | Ref | |
| yes | 194(31.6) | | 0.961(0.619-1.493) | 0.861 |
| unknown | 226(36.8) | | 1.068(0.681-1.675) | 0.776 |
| Quinolone therapy | | 0.596 | | 0.248 |
| no | 145(23.6) | | Ref | |
| yes | 469(76.4) | | 1.313(0.827-2.087) | |
| Penicillin therapy | | 0.847 | | 0.674 |
| no | 592(96.4) | | Ref | |
| yes | 22(3.6) | | 0.798(0.280-2.279) | |

*(Continued)*

**Table 1.** (Continued)

| Factors | N(%)/x±s | Univariate | Multivariate | |
|---|---|---|---|---|
| | | P value | HR(95%CI) | P value |
| Cephalosporin therapy | | 0.893 | | 0.124 |
| no | 456(74.3) | | Ref | |
| yes | 158(25.7) | | 1.368(0.918-2.039) | |
| Macrolide therapy | | 0.874 | | 0.449 |
| no | 586(95.4) | | Ref | |
| yes | 28(4.6) | | 1.341(0.628-2.865) | |
| Nitroimidazole therapy | | 0.985 | | 0.221 |
| no | 589(95.9) | | Ref | |
| yes | 25(4.1) | | 1.722(0.721-4.111) | |
| Antitubercular therapy | | <0.001 | | <0.001 |
| no | 377(61.4) | | Ref | |
| yes | 237(38.6) | | 0.394(0.264-0.588) | |
| Oral prednisone | | <0.001 | | <0.001 |
| no | 245(39.9) | | Ref | |
| yes | 369(60.1) | | 2.534(1.615-3.975) | |
| Bromocriptine therapy | | 0.371 | | 0.553 |
| no | 344(56.0) | | Ref | |
| yes | 270(44.0) | | 1.137(0.745-1.735) | |
| Abscess drainage | | 0.005 | | 0.011 |
| no | 186(30.3) | | Ref | |
| yes | 428(69.7) | | 0.506(0.300-0.854) | |
| Surgical excision | | 0.074 | | 0.022 |
| no | 128(20.8) | | Ref | |
| yes | 486(79.2) | | 0.585(0.369-0.927) | |

Abbreviations: HR, hazard ratio; CI, confidence interval; Ref, reference. The normal reference range for serum prolactin is 80.56–650.84 uIU/ml.

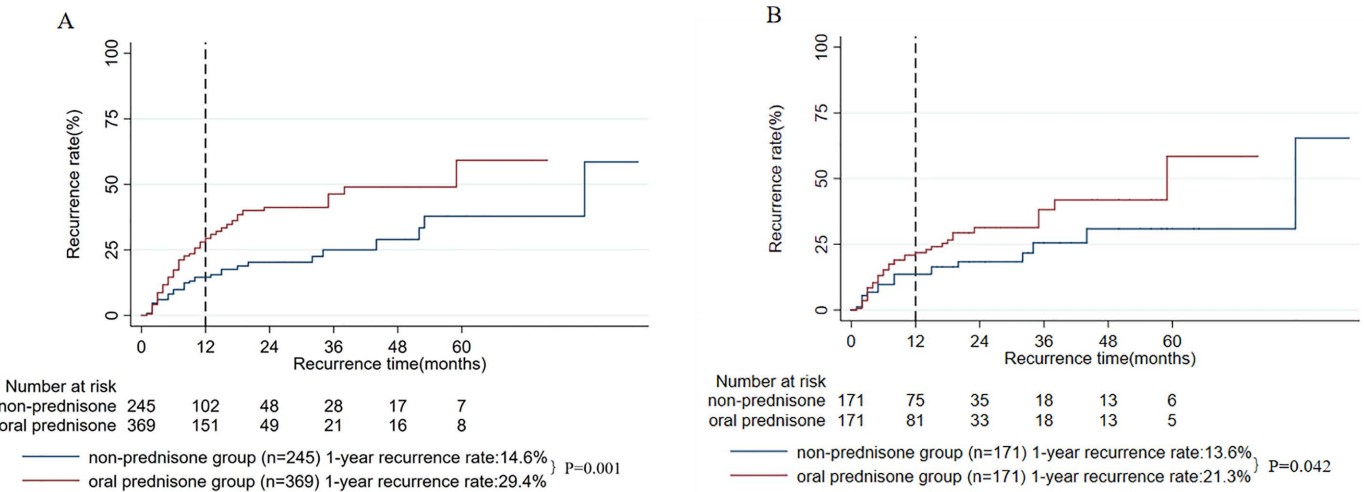

**Fig 2. A: Comparison of 1-year recurrence rates: oral prednisone versus non-prednisone groups before propensity score matching; B: Comparison of 1-year recurrence rates: oral prednisone versus non-prednisone groups after propensity score matching.**

171 matched pairs (all P > 0.05 after matching, S2 Table). In this matched cohort, the oral prednisone group continued to show a significantly elevated 1-year recurrence rate compared to the non-prednisone group (21.8% vs. 13.6%, P = 0.042, Fig 2B). These results indicate that oral prednisone exposure is independently associated with an increased risk of recurrence, even after accounting for baseline differences through rigorous matching.

## Treatment timing

To further investigate the impact of the timing of oral prednisone therapy on disease recurrence, we assessed the association between the time interval from symptom onset to treatment initiation and clinical outcomes. Of the 369 eligible patients, 335 with complete data on the symptom-to-treatment interval (34 missing, 9.2%; mean: 5.2 weeks; SD: 7.1 weeks) were analyzed. Using X-tile software, the optimal cutoff for this interval was identified as 6 weeks. Stratifying the cohort accordingly, we found that patients in the short-interval group (≤ 6 weeks, n = 253) had a 1-year recurrence rate of 33.5%, which was significantly higher than the 15.7% observed in the long-interval group (> 6 weeks, n = 82) (P = 0.016, Fig 3A). The time-varying recurrence analysis further confirmed the 6-week cutoff, showing that the 1-year recurrence rate peaked (range: 29.1%−38.4%) when treatment started within 6 weeks but plummeted to 15.1% and 14.4% for initiations at 7–8 and ≥ 9 weeks, respectively (Fig 3B).

## Treatment duration

We assessed the effect of oral prednisone treatment duration (excluding maintenance doses) on outcomes in a cohort of 341 patients with available data (from an original cohort of 369; 28 excluded due to missing exposure data, 7.6%). The mean treatment duration was 3.5 weeks (SD: 2.7 weeks). X-tile software determined 4 weeks as the optimal cutoff for recurrence risk stratification. Stratification of the cohort based on this cutoff revealed no significant difference in the 1-year recurrence rate between the short-duration (< 4 weeks, n = 215) and long-duration (≥ 4 weeks, n = 126) groups (30.0% vs. 30.5%, P = 0.381, Fig 4A). Time-dependent analysis (Fig 4B) further validated that the 1-year recurrence rate was similar across various treatment durations, ranging from 23.7% (0–1 week) to 33.0% (≥ 6 weeks), which suggests that the length of oral prednisone course (excluding maintenance) was not a major determinant of recurrence.

## Treatment dosage

The investigation of the impact of the initial dose of oral prednisone on prognosis comprised two parts: the absolute initial dose (mg/day) and the weight-adjusted dose (mg/kg/day). First, we analyzed the absolute initial dose. From the original cohort, 337 patients with complete records were included (32 excluded due to missing data, 8.7%). The mean absolute dosage of oral prednisone was 24.7 ± 7.7 mg. Using X-tile software, the optimal cutoff was determined to be 20 mg/day. Patients were stratified into low-absolute-dose group (≤ 20 mg/day, n = 163) and high-absolute-dose group (> 20 mg/day, n = 174). The analysis revealed a significantly lower 1-year recurrence rate in the low-absolute-dose group compared to the high-absolute-dose group (24.4% vs. 35.6%, P = 0.031, Fig 5A). Time-dependent analysis (Fig 5B) revealed a critical threshold: recurrence rates showed a notable jump starting at the 25–30 mg/day dosage (33.1%), with the sharpest increase observed in the ≥ 35 mg/day group (43.2%). This is in contrast to the lower rates seen at ≤ 10 mg/day (26.8%) and 15–20 mg/day (24.1%), highlighting a significantly elevated risk beyond 25 mg/day.

Second, we assessed the impact of the weight-adjusted dose (mg/kg/day) of oral prednisone on prognosis. From the original cohort, 337 patients with complete records were included in this analysis (32 were excluded due to missing data, 8.7%). The mean weight-adjusted dose was 0.41 ± 0.12 mg/kg/day. Using X-tile software, the optimal cutoff value was determined to be 0.43 mg/kg/day. Patients were stratified into low weight-adjusted dose group (≤ 0.43 mg/kg/day, n = 202) and high weight-adjusted dose group (> 0.43 mg/kg/day, n = 135). The analysis showed that the low weight-adjusted dose group had a significantly lower 1-year recurrence rate compared to the high weight-adjusted dose group (25.4% vs. 37.3%, P = 0.038, Fig 6A). The 337 patients were categorized into five groups according to the weight-adjusted

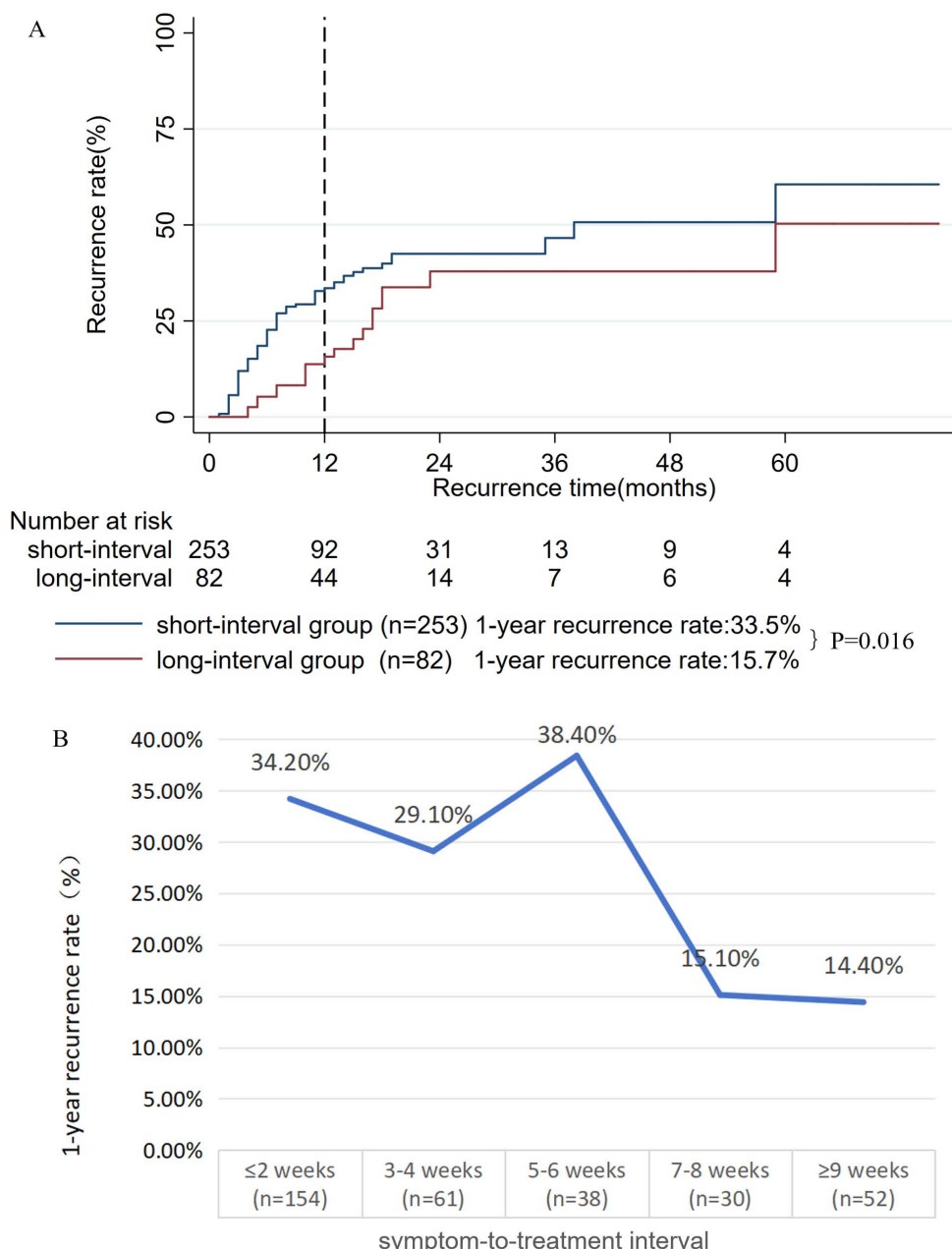

Fig 3. A: Kaplan-meier survival curves for the short-interval and long-interval groups; B: Sharp decline in 1-year recurrence rate with treatment delay beyond 6 weeks.

prednisone dose (mg/kg/day), as follows (**Fig 6B**): 0.2 mg/kg/day (≤ 0.24), 0.3 mg/kg/day (0.25–0.34), 0.4 mg/kg/day (0.35–0.44), 0.5 mg/kg/day (0.45–0.54), and 0.6 mg/kg/day (≥ 0.55). Analysis of the dose-response relationship revealed a non-linear trend in the one-year relapse rate. The rate was lowest (22.5%) at 0.3 mg/kg/day, increased modestly at 0.4 mg/kg/day (27.1%), and then rose sharply to 38.9% at 0.5 mg/kg/day, remaining elevated at 0.6 mg/kg/day (34.3%). Since the risk of relapse increased significantly beyond the 0.4 mg/kg/day dose, this level was proposed as the optimal cutoff value for maximizing efficacy and minimizing risk.

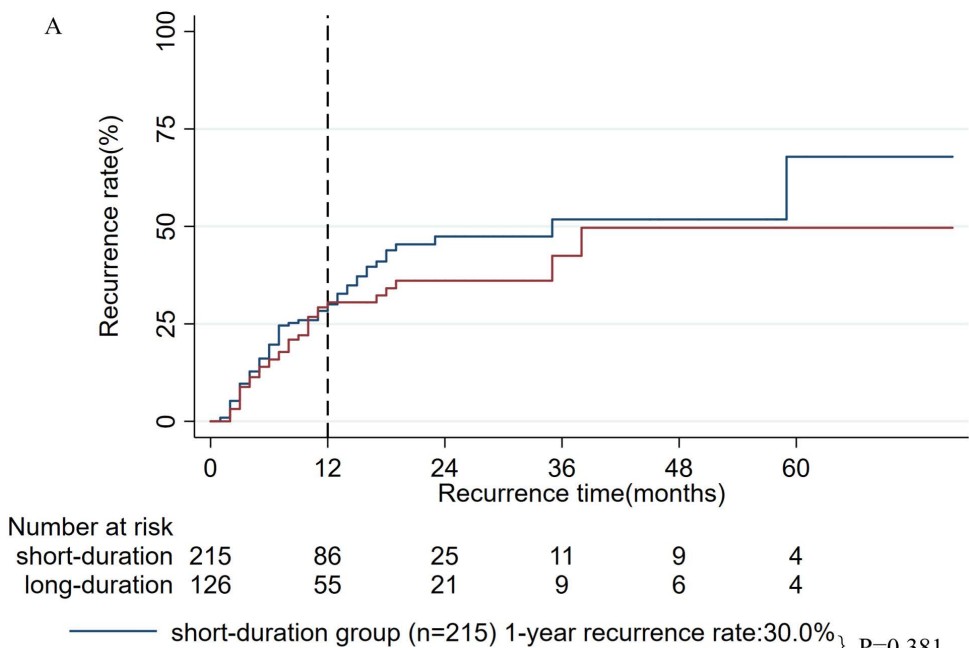

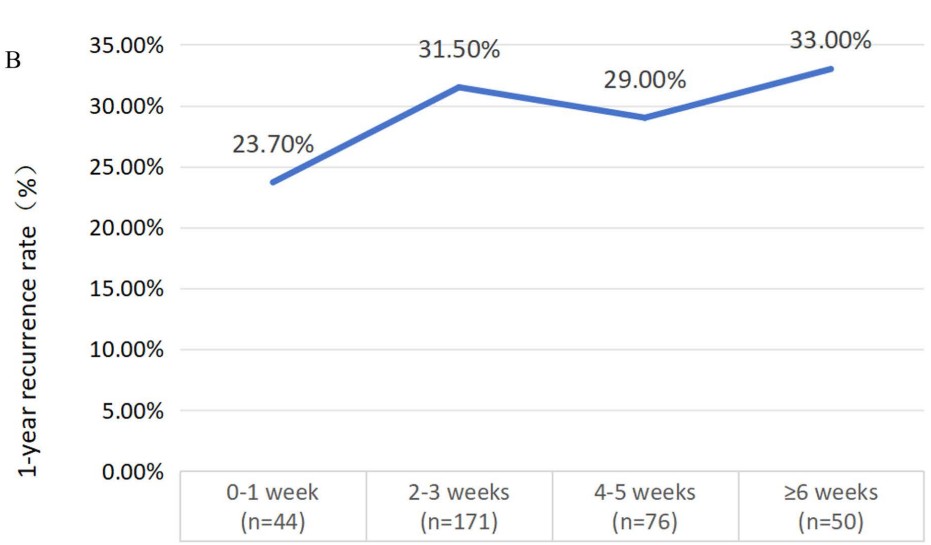

**Fig 4. A: Kaplan-meier survival curves for the short-duration and long-duration groups; B: The 1-year recurrence rate by oral prednisone duration.**

## Subgroup analysis

Based on these findings, we analyzed 306 patients who received oral prednisone treatment. The patients were initially stratified into Groups 1–8 (**Table 2**) according to the symptom-to-treatment interval, weight-adjusted dose (mg/kg/day), and treatment duration (63 patients were excluded due to missing essential grouping data, from an original cohort of 369

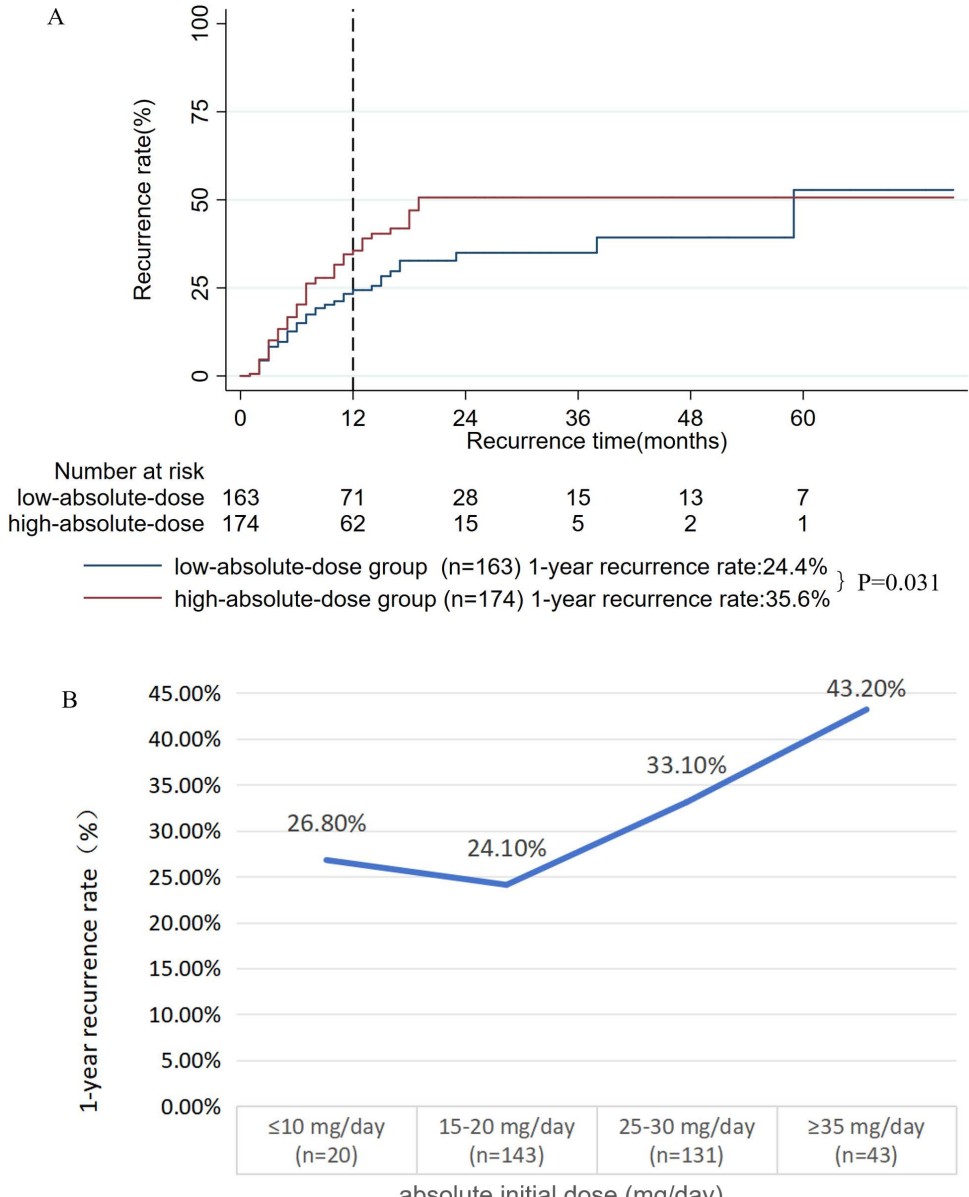

**Fig 5. A:** Kaplan-meier survival curves for the low-absolute-dose and high-absolute-dose groups; **B:** Time-dependent analysis of 1-year recurrence rate by absolute initial oral prednisone dosage.

patients). The weight-adjusted dose (mg/kg/day) was selected over the absolute initial dose (mg/day) because corticosteroid dosing is typically adjusted according to body weight in clinical practice. Survival analysis showed that among patients with a symptom-to-treatment interval of 0–6 weeks (Groups 1–4), the one-year relapse rates ranged from 30.9% to 37.6%, with no statistically significant differences between these groups (all $P > 0.05$). However, Groups 1–4 had significantly higher relapse rates than Group 5 (3.8%, all $P < 0.05$, **Table 2**). Therefore, patients from Groups 1–4 were merged with 22 additional patients who had a symptom-to-treatment interval of 0–6 weeks but lacked complete data on dose or treatment duration, forming the short-interval group. Among patients with a symptom-to-treatment interval of 0–6

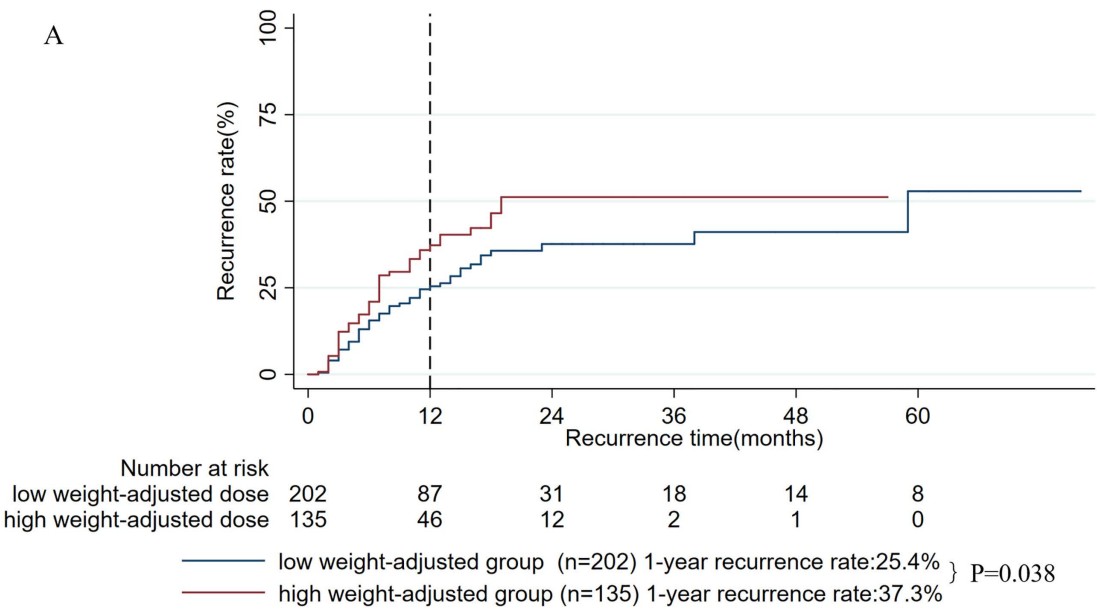

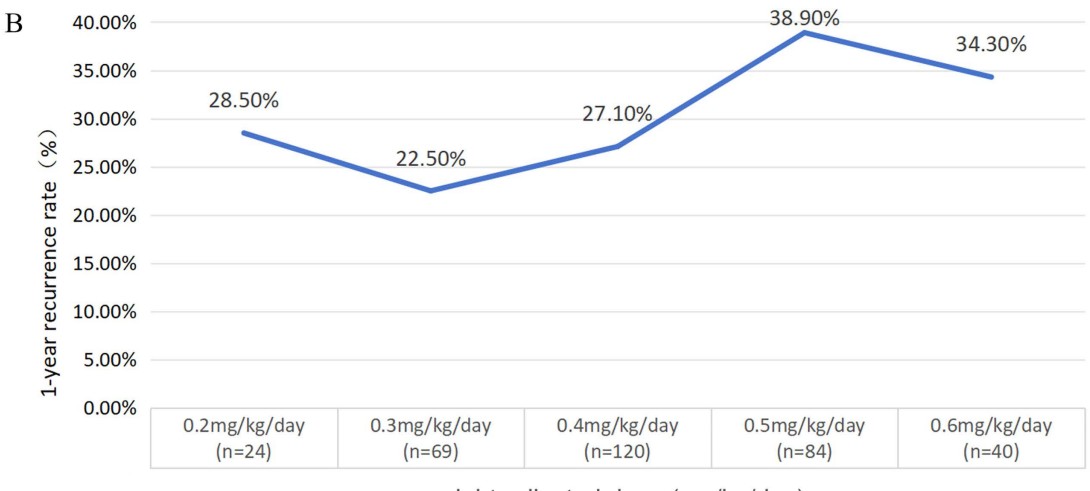

Fig 6. A: Kaplan-meier survival curves for the low weight-adjusted dose and high weight-adjusted dose groups; B: Identification of a critical risk threshold at 0.4 mg/kg/day for weight-adjusted oral prednisone dose using time-dependent analysis.

weeks and a dosage > 0.4 mg/kg/day (Groups 7 and 8), the one-year relapse rates were comparable (26.9% vs. 28.6%, P = 0.289, **Table 2**). Thus, Groups 7 and 8 were combined with 1 other patient meeting the same clinical criteria (symptom-to-treatment interval of 0–6 weeks and dosage > 0.4 mg/kg/day) but with missing treatment duration data, to constitute short interval-high dose group. For patients with a symptom-to-treatment interval of 0–6 weeks and a dosage ≤ 0.4 mg/kg/day (Groups 5 and 6), the one-year relapse rates were also not significantly different (3.5% vs. 11.1%, P = 1.000, **Table 2**). Accordingly, short interval-low dose group was formed by merging Group 5, Group 6, and an additional 2 patients who had a symptom-to-treatment interval of 0–6 weeks and a dosage ≤ 0.4 mg/kg/day, but whose treatment duration data were missing.

**Table 2. The difference in 1-year recurrence rates among groups 1-8.**

| Group | Symptom-to-treatment interval (weeks) | Dosage (mg/kg/day) | Duration (weeks) | N | 1-year recurrence rate (%) | P value | P value | P value | P value | P value | P value | P value |
|---|---|---|---|---|---|---|---|---|---|---|---|---|
| 1 | 0-6 | ≤0.4 | 0-3 | 60 | 34.5 | Ref | | | | | | |
| 2 | 0-6 | ≤0.4 | ≥4 | 53 | 34.8 | 0.705 | Ref | | | | | |
| 3 | 0-6 | >0.4 | 0-3 | 84 | 37.6 | 0.521 | 0.328 | Ref | | | | |
| 4 | 0-6 | >0.4 | ≥4 | 34 | 30.9 | 0.756 | 0.928 | 0.408 | Ref | | | |
| 5 | ≥7 | ≤0.4 | 0-3 | 27 | 3.8 | 0.009 | 0.030 | 0.004 | 0.026 | Ref | | |
| 6 | ≥7 | ≤0.4 | ≥4 | 12 | 11.1 | 0.075 | 0.145 | 0.049 | 0.114 | 1.000 | Ref | |
| 7 | ≥7 | >0.4 | 0-3 | 23 | 26.9 | 0.348 | 0.669 | 0.176 | 0.556 | 0.033 | 0.125 | Ref |
| 8 | ≥7 | >0.4 | ≥4 | 13 | 28.6 | 0.125 | 0.217 | 0.079 | 0.149 | 0.694 | 0.609 | 0.289 |

Abbreviations: Ref, reference.

To further investigate the impact of oral prednisone administration on prognosis, a subgroup analysis was performed. The analysis revealed a clear hierarchy in relapse risk: the short-interval group had the highest 1-year relapse rate (33.5%), which was significantly greater than that of the non-prednisone group (14.6%, P < 0.001, **Table 3**, **Fig 7**). In contrast, the relapse rate in the long interval-high dose group (25.9%) was comparable to that of the short interval group (P = 0.103, **Table 3**, **Fig 7**). Notably, the long interval-low dose group exhibited the most favorable outcome, with a relapse rate (6.2%) that was not only significantly lower than other dosing groups (all P < 0.05, **Table 3**, **Fig 7**) but also comparable to the non-prednisone group (P = 0.233, **Table 3**, **Fig 7**). These results underscore that a short dosing interval is associated with the highest risk of relapse.

## Discussion

Since its initial description in 1972 [23], no universally accepted treatment consensus has been established for GLM. The most widely accepted hypothesis posits GLM as an autoimmune process, which forms the pathophysiological rationale for employing corticosteroid therapy [24]. Studies have indicated that an initial course of oral steroids can significantly reduce breast and skin lesions, thereby potentially avoiding repeated and deforming operations [8]. To our knowledge, no study had previously provided a detailed, comparative analysis of oral prednisone treatment protocols for GLM, a critical knowledge gap this research sought to address.

Based on a relatively large cohort, this study observed that treatment with oral prednisone was associated with a significantly increased risk of recurrence in patients with GLM (HR: 2.534), which corresponded to an approximately 1.5-fold increase in recurrence risk. Although initial use of oral steroids could lead to noticeable regression of breast and skin lesions and facilitate more limited surgery [8,25], our findings suggested that this did not necessarily lead to a reduction

**Table 3. Comparison of 1-year recurrence rates among subgroups.**

| Group | Symptom-to-treatment interval (weeks) | Dosage (mg/kg/day) | Duration (weeks) | N | 1-year recurrence rate (%) | P value | P value | P value |
|---|---|---|---|---|---|---|---|---|
| non-prednisone group | — | | | 245 | 14.6 | Ref | | |
| short-interval group | 0-6 | no restriction | no restriction | 253 | 33.5 | <0.001 | Ref | |
| long interval-high dose group | ≥7 | >0.4 | no restriction | 37 | 25.9 | 0.467 | 0.103 | Ref |
| long interval-low dose group | ≥7 | ≤0.4 | no restriction | 41 | 6.2 | 0.233 | 0.002 | 0.038 |

Abbreviations: Ref, reference.

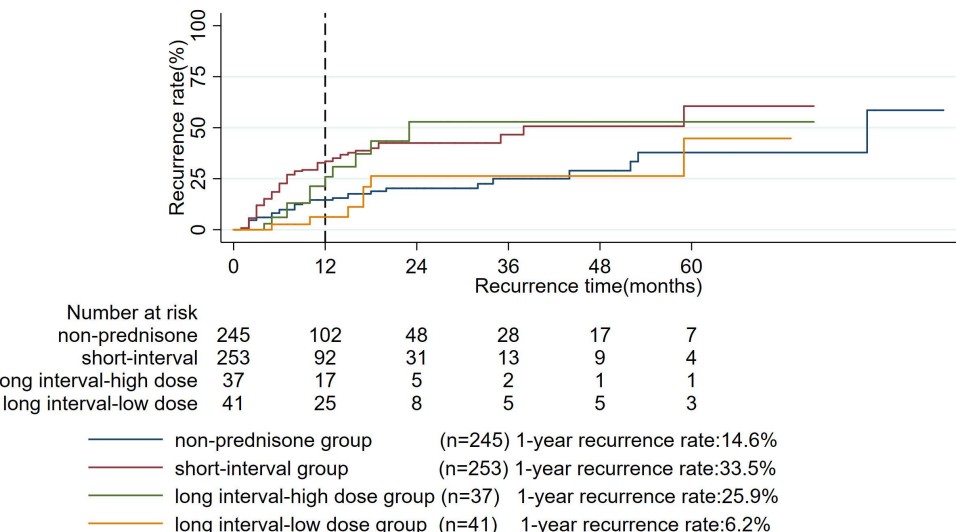

**Fig 7. Kaplan-meier survival curves for subgroups.**

in long-term recurrence rates. Several lines of evidence supported this conclusion. First, the 1-year recurrence rate with oral prednisone therapy in our cohort was 29.4%, a result that aligned with the wide range of rates (24.7% to 46.9%) reported in earlier studies [4,7,12,13,16,26,1]. In addition, while corticosteroid therapy was found to suppress inflammation and reduce lesion size by inhibiting pro-inflammatory cytokines, it seldom led to complete disease eradication [12]. The high recurrence rate following oral steroid therapy was thus thought to be due to the persistence of subclinical residual foci. Furthermore, several studies reported a higher recurrence rate with oral steroids compared with observation alone [16,26,27], as illustrated in a 2024 meta-analysis of 65 studies where recurrence rates were 24% and 11%, respectively [26]. These consistent findings across studies, including ours, strengthen the observed association, although its interpretation must consider the potential for confounding inherent in observational data. This association remained robust even after rigorously controlling for baseline confounders via PSM, which yielded a significantly higher recurrence rate in the matched prednisone group (21.8% vs. 13.6%). A notable aspect of this study was its sample size. To our knowledge, the cohort included more than 600 patients, which exceeded the size of most previous related studies [1,4,7,12–17,25,28–32] (typically involving ≤ 200 patients), thereby offering a more robust foundation for the evidence in this area, though causal inference is constrained by the non-randomized study design.

Although oral prednisone was widely used for GLM, the optimal timing for its initiation remained unclear. In this context, our findings highlighted the potential importance of treatment timing: a noticeable association with recurrence risk was observed, as patients treated within 6 weeks of onset experienced a recurrence rate exceeding 30%, compared to approximately 15% when treatment was initiated later (**Fig 3A**). In interpreting the observed association between early treatment initiation and higher recurrence, it is crucial to first consider a key alternative explanation: confounding by disease severity. Patients who presented and required treatment within the first 6 weeks likely had more active or severe disease at baseline than those who presented later. Thus, the higher recurrence in the early-treatment group may substantially reflect this greater intrinsic disease burden rather than a direct, negative effect of early treatment per se. Residual confounding by unmeasured severity factors cannot be entirely excluded and represents a primary limitation in causal inference. Even considering this predominant confounding effect, the interplay between the natural history of GLM and the immunomodulatory action of prednisone may offer complementary insights. First, if early immunosuppression during the acute phase achieves only temporary control without resolving underlying immune dysregulation, recurrence upon dose

reduction ("steroid tapering rebound") may be more likely. Second, the 6-week interval may act as a natural filter, allowing mild, self-limiting cases to resolve, thereby selecting a subgroup more amenable to systemic therapy. Third, delayed administration during a more chronic phase might modulate tissue repair more effectively, reducing pro-inflammatory scar formation that could serve as a nidus for recurrence. Collectively, these findings suggested that for GLM patients, there may be a preferable therapeutic window for oral prednisone—initiating treatment beyond 6 weeks, rather than very early, might be associated with improved long-term outcomes. This association requires further validation in studies designed to better control for baseline disease severity. To our knowledge, this study was among the first to systematically explore the timing of corticosteroid therapy in GLM, offering insights that could inform clinical practice.

No unified standard currently exists for the course of oral prednisone, and clinical practice often relies on empirical medication. This study found that the length of the oral prednisone course (excluding the maintenance phase) was not a major determinant of recurrence, a finding contrasting with reports from other study favoring longer fixed courses [25]. Several factors may explain these discrepancies. Firstly, the differential requirements for prednisone course length based on disease severity and treatment response—with longer courses needed for severe or refractory cases and shorter ones sufficient for mild or rapidly responsive cases—may have led to the inconsistent study results. Thus, treatment duration may act more as a marker of disease behavior than as an independent modifiable factor. Moreover, the specific treatment regimens likely differed in key aspects beyond just course length. Notably, the initial dose in our cohort (0.41 mg/kg/d) was substantially lower than the 0.75 mg/kg/d used in the study advocating longer courses [25], suggesting dose intensity may be more critical than duration alone. Finally, the treatment context is paramount. The preferred strategy combines preoperative steroids with surgery [8,33]. In contrast, prednisone monotherapy, while a necessary alternative for non-surgical candidates, is associated with higher recurrence risk and often requires prolonged therapeutic courses [16,26,27]. Our study included a subset of patients who did not undergo surgery. These patients likely needed extended steroid courses yet still experienced high recurrence rates. This observation—that prolonged monotherapy signifies a refractory, high-risk subgroup—explains why the highest recurrence rate (33.0%) in our cohort was found in patients with treatment courses ≥ 6 weeks. Consequently, an isolated comparison of "treatment duration" lacks informativeness without critical context, particularly regarding concomitant surgical resection. In this context, a key contribution of our study is its provision of quantitative evidence from a large-scale cohort supporting the potential efficacy of a lower-dose regimen. Future work is necessary to establish standardized protocols that define the role and timing of surgery in conjunction with steroid therapy for more definitive clinical guidance.

The dosage of oral prednisone was considered theoretically important for determining prognosis in GLM. However, the management of GLM with corticosteroids continued to be guided largely by clinical experience. Although current guidelines recommended a prednisone dose of 0.75 mg/kg/d [18], this full dosage was seldom achieved in clinical practice or research due to its magnitude. Importantly, few studies evaluated the impact of different dosages on clinical outcomes in GLM patients, leaving a notable evidence gap worldwide. In this study, which sought to address this question, a clear dose-outcome relationship was identified. The relapse risk increased notably at doses exceeding 0.4 mg/kg/d, suggesting that ≤ 0.4 mg/kg/day may serve as a reasonable therapeutic cutoff for balancing efficacy and safety. Crucially, when interpreting this relationship, an important consideration is that the prescribed dose may be correlated with disease severity at baseline (i.e., confounding by indication). It is plausible that patients with milder disease were prescribed lower doses, which could partly explain their better outcomes. As this study did not formally compare clinical severity between dose groups, this possibility cannot be ruled out and represents a key consideration for the results, also applying to the weight-adjusted dose analysis. Several previous studies supported the use of lower-dose prednisolone, reporting favorable outcomes and reduced recurrence rates [29,31]. One study demonstrated that a regimen of low-dose prednisolone (30 mg/day) combined with drainage was more effective and resulted in a lower recurrence rate than either surgical resection alone or high-dose prednisolone (50 mg) [29]. Similarly, Shin et al. [31] reported that initiating treatment with a steroid dose of 20 mg/day, in combination with surgery, was associated with a recurrence rate of

only 7.1%. Together, these findings suggested that lower-dose corticosteroid regimens, particularly when integrated with surgical intervention, may help reduce recurrence. However, an alternative perspective was provided by Montazer et al. [32], who reported that a high-dose prednisone regimen (50 mg/day) was associated with better success rates and lower recurrence compared to a low-dose regimen (5 mg/day), potentially reducing the need for surgery. Despite these contrasting findings, the results of the present study were supported by several considerations. First, the analysis was based on a relatively large cohort [1,7,12–17,25,28–32], which helped improve the reliability of the conclusions. Nevertheless, the potential confounding effect of disease severity on dose assignment persists. Second, the rationale for a lower-dose strategy aligned with the principle of using the minimal effective dose to avoid excessive immunosuppression—a consideration especially relevant given the self-limiting nature of some GLM cases. This approach was also consistent with the Chinese expert consensus, which recommended an initial methylprednisolone dose of 20 mg [18,19]. Third, higher-dose regimens were often associated with more pronounced side effects, which could affect treatment adherence and thus influence outcomes [14], further supporting the exploration of an optimal, lower-dose strategy. To our knowledge, this was among the first studies to systematically examine the relationship between prednisone dosage and clinical outcomes in GLM. Addressing this evidence gap represented a useful step toward more informed treatment guidance. It is important to contextualize these findings within the observational nature of our study. The specific protocols for initiating and tapering prednisone were not uniform but reflected prevailing clinical practice. The absence of standardized definitions for an "initial dose" and a consensus tapering schedule highlights a significant gap in current GLM management. Future research must move toward prospective designs that define and test specific treatment protocols to establish a robust evidence base for optimal steroid use in GLM.

Based on subgroup analyses, a clear hierarchy in relapse risk emerged. The long-interval, low-dose group had the most favorable outcome (1-year relapse rate: 6.2%), comparable to the non-prednisone group. In contrast, groups initiating treatment within 6 weeks had consistently high relapse rates (30.9%−37.6%). Integrating these findings, we propose the following clinical guidance to optimize the use of oral prednisone in GLM: (1) Timing and dose are paramount. If prednisone is used, initiating therapy beyond 6 weeks after symptom onset at a lower dose (≤ 0.4 mg/kg/day) is associated with the lowest relapse risk. (2) Avoid unnecessary exposure. For patients with small lesions and no marked systemic inflammatory response, the use of prednisone should generally be avoided. (3) Prioritize surgery in complex cases. For patients presenting with complex abscesses or fistulas, the cornerstone of management should be timely surgical intervention. Prednisone, if used, should play only a limited adjunctive role. It should be noted that these subgroup analyses regarding treatment timing, dosage, and duration are exploratory in nature. The compared subgroups (e.g., short- vs. long-interval, low- vs. high-dose) were not randomized and may have inherent baseline differences. Therefore, these findings, while informative and hypothesis-generating, require further validation in future prospective, ideally randomized, studies to confirm the proposed optimal treatment window and dosing strategy.

This study has several limitations that should be acknowledged. First, its single-center, observational design may limit generalizability and introduces the potential for unmeasured confounding (e.g., confounding by indication for prednisone use), despite adjustments for multiple severity indicators. Specifically, while propensity score matching was employed to balance observed confounders including concomitant therapies, it cannot account for unmeasured or residual confounding factors. Furthermore, the reduction in sample size after matching may affect statistical power and the generalizability of the matched cohort. Second, although we applied a standardized protocol, the hybrid retrospective-prospective design over a long timeframe (2017−2024) carries an inherent risk of subtle inconsistencies in data or practice. Third, the analyses of specific prednisone parameters (timing, dose, duration) were exploratory and based on complete-case data; the missingness, though low (7.6%−9.2%), was not imputed. Fourth, and most pertinent to the primary findings regarding treatment parameters, the optimal thresholds for timing and dose were identified using the data-driven X-tile software. Although these thresholds were subsequently validated via time-trend analysis, multi-dimensional subgroup analysis, and multivariable modeling, the inherent risk of overfitting from such methods cannot be entirely excluded. These findings

provide important hypotheses for optimizing prednisone strategies, and the precise cutoff values warrant external validation in prospective studies. Finally, this study focused on isolated prednisone use; defining its optimal role in combination with surgery remains a critical question for future research.

## Conclusions

The findings suggested that oral prednisone might be associated with increased relapse rates in GLM patients. The analysis indicated that the symptom-to-treatment interval appeared to be the most influential factor in treatment outcomes, followed by weight-adjusted dosage, while treatment duration did not show a significant effect. This supports a cautious approach to prednisone use in GLM management. Based on these observations, a treatment approach involving prednisone initiation after a 6-week symptom interval, at a dose of ≤ 0.4 mg/kg/day, may be considered. Further studies will be needed to validate these preliminary findings and refine the proposed strategy.

## Supporting information

**S1 Table. Univariable and multivariable Cox regression analysis restricted to the prospective cohort (n = 374).**
(DOCX)

**S2 Table. Baseline characteristics of patients before and after propensity score matching.**
(DOCX)

**S1 Checklist. PLOSOne clinical studies checklist.**
(DOCX)

## Author contributions

**Conceptualization:** Haiyan Zhang, Jing Luo.

**Data curation:** Ruiyang Wu, Jin Chen.

**Formal analysis:** Haiyan Zhang, Ruiyang Wu, Jin Chen.

**Investigation:** Jinyan Feng, Jing Luo.

**Methodology:** Ruiyang Wu, Jin Chen, Jinyan Feng.

**Supervision:** Haiyan Zhang, Jing Luo.

**Validation:** Haiyan Zhang, Ruiyang Wu.

**Writing – original draft:** Haiyan Zhang, Ruiyang Wu, Jin Chen.

**Writing – review & editing:** Haiyan Zhang, Jing Luo.

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
