## [Decision Letter · Decision Letter 0]

12 Dec 2025

Dear Dr. Luo,

Thank you for submitting your manuscript to PLOS ONE. After careful consideration, we feel that it has merit but does not fully meet PLOS ONE’s publication criteria as it currently stands. Therefore, we invite you to submit a revised version of the manuscript that addresses the points raised during the review process.

We look forward to receiving your revised manuscript.

Kind regards,

Siddharth Gosavi, MBBS, MD Internal Medicine,DNB Internal Medicine

Academic Editor

PLOS One

2. In the online submission form you indicate that your data is not available for proprietary reasons and have provided a contact point for accessing this data. Please note that your current contact point is a co-author on this manuscript. According to our Data Policy, the contact point must not be an author on the manuscript and must be an institutional contact, ideally not an individual. Please revise your data statement to a non-author institutional point of contact, such as a data access or ethics committee, and send this to us via return email. Please also include contact information for the third party organization, and please include the full citation of where the data can be found.

Additional Editor Comments:

This large-scale retrospective cohort study, which examined 614 patients diagnosed with GLM between 2017 and 2024, evaluated the effect of prednisolone on relapse in terms of treatment start time, duration, and starting dose. Analysis showed that oral prednisolone use was an independent risk factor for relapse in GLM (HR 2.534). The one-year relapse rate was significantly higher in patients given prednisolone (29.4% vs. 14.6%). However, subgroup analyses showed that starting treatment within 6 weeks of symptom onset and using a dose of ≤0.4 mg/kg/day significantly reduced relapse rates (1-year relapse 6.2%). This rate was similar to the group not receiving prednisolone. In contrast, early treatment initiation (0–6 weeks) was associated with the highest relapse rate (33.5%). In conclusion, although prednisolone generally increases the risk of relapse in GLM, delayed start and low-dose protocols appear to be the most appropriate treatment strategy to reduce this risk.

İntro:

The introduction provides a well-positioned, logical scientific basis. Clarify the purpose in the final section, aligning it with the study design.

MM&Results

Major Comments

The manuscript combines retrospective data (2017–2022) with prospectively collected data (from 2022 onward). This hybrid design introduces potential inconsistencies in data completeness and diagnostic/treatment protocols across time periods. The authors should clarify:

*how data collection was standardized across the retrospective and prospective phases,

*whether the two periods differ significantly in baseline characteristics or treatment strategies,

*whether a sensitivity analysis excluding early retrospective years was performed.

Prednisone administration was not randomized and may reflect clinician judgment based on disease severity. This introduces substantial bias:

*severe cases may have been preferentially treated earlier or with higher doses,

*the higher recurrence rate in the prednisone group may therefore reflect baseline severity rather than treatment effect.

Different analyses include different patient counts (e.g., 335 for timing analysis, 341 for duration, 337 for dose). The mechanism of missing data (MCAR/MAR/MNAR) is not described, and there is no imputation strategy. Given that missingness may introduce bias, the authors should:

*report missing data percentages for each variable,

*describe the missing data mechanism,

*and consider multiple imputation if appropriate.

The manuscript uses X-tile to identify cutoffs for treatment timing, dose, and duration. This method increases the risk of overfitting and inflated type I error, especially without an external validation cohort. The authors should clarify:

*whether correction for multiple testing was applied,

*whether cutoffs were validated using bootstrap or internal cross-validation,

*and how robust these thresholds are across sensitivity analyses.

Patients received diverse additional therapies, including antimicrobials, anti-tuberculosis agents, drainage procedures, and bromocriptine. Many of these are likely to influence recurrence risk. The multivariable Cox model includes some of these factors, but:

*their distribution between exposed and unexposed groups is not reported,

*no stratified or interaction analyses are provided,

*the possibility of treatment clustering (e.g., prednisone + drainage) is not explored.

*A more granular adjustment or subgroup analysis is warranted.

“Return of symptoms after clinical improvement or cure” is somewhat subjective and operator-dependent. Since recurrence is the primary endpoint, the authors should:

*specify objective criteria (e.g., imaging, physical examination, laboratory data),

*clarify whether recurrence was assessed by blinded observers,

*and report inter-observer variability if applicable.

Some subgroups used in the final survival analyses appear very small due to filtering (e.g., n=1 or n=2 cases added). This reduces reliability of hazard estimates. The authors should:

*report subgroup sample sizes clearly,

*refrain from overly granular stratification when numbers are insufficient,

*discuss the limitations in the power of subgroup comparisons.

Minor Comments

The dosing description (“initiated at a high dose and tapered by 5 mg every 3–7 days”) lacks:

*the definition of “high dose,”

*typical taper duration,

*criteria for tapering speed.

*A standardized protocol description is needed for reproducibility.

Figure 1 outlines the workflow but does not provide a CONSORT-style patient flow diagram. A flowchart of inclusions/exclusions and analysis subsets would greatly improve transparency.

The manuscript lacks a comparison of baseline characteristics between:

*prednisone vs. non-prednisone groups,

*short- vs. long-interval groups,

*low- vs. high-dose groups.

Rationale for Dividing Dosage by Both Absolute and Weight-Adjusted Methods: Both approaches are analyzed, but the rationale and clinical relevance should be explained more succinctly.

Language: The Methods section is thorough but dense. Some subsections may benefit from condensation for readability.

In summary;

This manuscript is strengthened by a large sample size and a comprehensive multidimensional analysis of prednisone therapy. However, concerns regarding confounding, missing data, exposure classification, and multiple threshold testing must be addressed before the conclusions can be fully supported. Clarification of the methodology and additional analyses as suggested above would considerably improve the manuscript’s rigor and impact.

Discussion:

The discussion is generally aligned with the study's three main objectives: treatment timing, dose, and duration; this is good.

However, although the reasons for choosing these three dimensions—the clinical relevance of the research gap—were strongly emphasized in the introduction, the discussion occasionally becomes disjointed. Some lengthy explanations (e.g., pathophysiology, steroid rebound mechanism) distract from the study's main question.

The comparison with the literature is comprehensive; many studies are compared. However, the comparisons are presented in very long passages, somewhat obscuring the clinical implications. In some sections, citations are listed consecutively, but the question "What does this study add to the literature?" is not clearly stated.

The proposed treatment strategy at the end of the discussion is a strong point; however, more targeted conclusions, such as for whom it should be applied and for which clinical phenotypes, would have added clinical value.

Limitations are accurately listed.

However, the heterogeneity of surgical interventions and the impact of nonsteroidal therapies, particularly as a critical limitation, should have been integrated into the discussion earlier.

IGM is a disease with many unknown issues, especially regarding its treatment. Studies including a large number of patients, like this one, could be very helpful in clarifying the concern of using steroids and their adverse effects in IGM.

The three dimensions considered for grouping treatment features and the cut-off point measurements are interesting, providing a well-designed analysis protocol. However, there are many concerns that need to be addressed before publication could be considered for this manuscript:

1. ABSTRACT: needs some English language editing. Also, the main concerns that will be approached below should be mentioned in brief in the abstract.

INTRODUCTION:

2. All through the introduction, the past tense is wrongly used for present data on GLM.

3. Lines 44 to 47: The authors mention that IGM incidence has rapidly increased during the present decade. For this, they cite a paper from 2019, which is a clinical study on the surgical treatment of GLM. That papers cite, for stating the incidence conditions, a paper from 2009, which reports a series of 54 cases of GLM. None of these investigates the incidence of GLM, and the year of publication of the latter is too old to talk about the recent decade. Therefore, this sentence is not supported by the present evidence. To update the issue, please know that the only papers published up to now (as far as I know) with an acceptable assessment of GLM incidence are those of the US CDC [Goldman M, et al. Idiopathic Granulomatous Mastitis in Hispanic Women--Indiana, 2006-2008. MMWR: Morbidity & Mortality Weekly Report. 2009 Dec 4;58(47)], which only includes Indiana (the city with the highest Hispanic population in the US) and cannot be applicated globally, and that of Zou et al [Zou J, et al. Clinical trends in granulomatous mastitis incidence, prevalence, and treatment: A retrospective study highlighting ethnic differences in care. Journal of Surgical Research. 2024 Oct 1;302:732-8.] which consists of a population-based study in the USA. The results of the latter are very valuable, but can only be applied to the US population, and they do not assess the recent incidence trends. So, the data in these lines of the manuscript lack evidence, unless the authors provide a correct reference.

4. Lines 55-57: ‘Although topical steroid therapy was reported as a useful treatment modality [6,11], its invasive nature limited its widespread clinical adoption.’ Topical steroids are the least invasive treatment, how do authors mention that the reason for its limited use is its invasive nature?

5. Lines 52-67: While using many references about it, the authors forgot to mention the recently adopted local (intralesional) use of steroids. May be, by topical use and its invasiveness, they meant local injections??

6. Introduction overall: The introduction is not well-written. Authors could have a better approach to the present therapeutic approaches and the existing challenges in GLM treatment. Also, despite the large amount of literature available, some references are selected inappropriately, like the use of references 1 (comparing local and oral steroids in GLM treatment) and 2 (review of GLM pathology) for the first sentence (‘ Granulomatous lobular mastitis was a rare, benign, and chronic inflammatory disease of the breast with an unknown etiology’).

METHODS

7. Lines 80-83: Before mentioning the number of excluded patients, the inclusion and exclusion criteria should be stated clearly.

8. Line 84: 1) What was the mode of diagnosis of GLM? What were the diagnostic criteria?2) Did the included patients used only prednisolone as their treatment modality? How did the authors get sure about the concomitant use of NSAIDs, or any combination treatment?

9. Please explain when were the lab tests done? As the test results were related to disease prognosis, the timing of the tests is very important. Usually, all GLM patients do not attend at the first disease presentation. Therefore, tests are taken at different disease stages. How was the condition of lab test timing in your database? If the mentioned concerns existed in your study (as expected), please mention this limitation in the discussion.

10. Fig.1: Although well-organized and very useful, I suggest that sentences are shortened. Usually, short sentences or expressions are used in these types of workflows. E.g., instead of ‘614 pts with GLM were ultimately included in the analysis’, a shortened version as ’Finally included: 614 patients’ would be more suitable for the figure. This should be done in other parts of the figure also. Importantly, the ’It was hypothesized that …’ is not appropriate for the figure. It could be replaced by something like ’Assessment of prednisolone effects’.

11. Line 144: How do authors explain the effect of the affected side on prognosis?

12. Tab 1: For prolactin, please provide the normal range. Also, define hyperlipidemia (which types of lipids? The ranges?). Was bromocriptine used for hyperprolactinemia? These should be described shortly in the footnote to make the table understandable and useful.

13. Tab. 1: For the age at diagnosis, which carried a worse prognosis, younger or elder age?

14. Lines 152-154: These descriptions should come in the Methods, not the results. Also, please explain what were the non-prednisone treatments. The overall treatment strategies are described in lines 97-100; however, we need to know, for example, if you had patients who only received antibiotics? Or only anti-tuberculosis agents? Or if bromocriptine was used alone, and regardless of the prolactin status? Regarding the higher recurrence in the prednisone group (as mentioned in lines 154-156 and Fig 2), it is crucial to know about the comparison group details.

DISCUSSION

15. Lines 283-285: The point is that the two groups: prednisone and non=prednisone are not compared regarding disease severity, so the results might just mirror the disease severity. Please explain if that is not the point, and otherwisw discuss the point as such and add it in the limitations.

16. Lines 309-327: These are hypothetical reasons for the higher recurrence in patients with early-onset treatment, and they do not follow a very sound rational. Patients who attended later probably had a less severe disease, and those that attended earlier had a more severe disease. So, treatment was begun sooner, but recurrence was higher. Although this is not certain, but it should be mentioned as a probable reason for these results, and recognized as a main confounding factor in the discussion and limitations.

17. Lines 328-331: Treatment duration depends on the disease response to treatment, so medicines cannot be cut before obtaining an acceptable response. Do the authors mean to discontinue steroids while the disease is improving but we are still waiting for recovery? The reason for the lower recurrence with a shorter treatment duration might simply be because of less severe disease. Authors should consider this important point in the discussion and limitations.

18. Lines 332 to 358: This part should be shortened and simplified with an accurate wording. Authors should pay attention to the tense of the verbs and the use of ‘this’ and ‘that’ for theirs vs. other studies.

19. Lines 367-369: Lower doses are most probably due to less severe disease, which explains the lower recurrence. As the clinical signs are not compared between the two groups, we cannot know whether the low and high dose groups were similar in this regard, and they were most probably not. This also applies to the weight-adjusted dose. This key point should be considered and alluded to in the discussion and limitations.

20. Apparently, the results and discussion indicate that the non-prednisone treatments, the lower doses of prednisone, the later-onset treatment initiation, and the shorter duration of prednisolone use is either better that or not different with the comparison group better. Wouldn’t all these reflect that prednisolone is better not used?

21. Overall, as disease severity has not been considered in the comparison between groups, results should be taken cautiously. The very positive sentences that present results of this study as doubtless data need to be re-written to reflect the uncertainty of the mentioned points. The limitations also should be revised accordingly.

Reviewers' comments:

Reviewer's Responses to Questions

**Comments to the Author**

1. Is the manuscript technically sound, and do the data support the conclusions?

Reviewer #1: Yes

Reviewer #2: Yes

Reviewer #3: Partly

2. Has the statistical analysis been performed appropriately and rigorously?

Reviewer #1: Yes

Reviewer #2: Yes

Reviewer #3: Yes

3. Have the authors made all data underlying the findings in their manuscript fully available?

Reviewer #1: Yes

Reviewer #2: No

Reviewer #3: Yes

4. Is the manuscript presented in an intelligible fashion and written in standard English?

Reviewer #1: Yes

Reviewer #2: Yes

Reviewer #3: Yes

Reviewer #1: The study is well designed and it will help to physicians to use oral steroid therapy for the patients with IGM. In fact I use intralesional steroid therappy in the treatment of IGM. However this study has investigated various therapies for the patients and explained the dosage and the timing of initiation of oral steroid therapy. So, it is available for publication.

Reviewer #2: This large-scale retrospective cohort study, which examined 614 patients diagnosed with GLM between 2017 and 2024, evaluated the effect of prednisolone on relapse in terms of treatment start time, duration, and starting dose. Analysis showed that oral prednisolone use was an independent risk factor for relapse in GLM (HR 2.534). The one-year relapse rate was significantly higher in patients given prednisolone (29.4% vs. 14.6%). However, subgroup analyses showed that starting treatment within 6 weeks of symptom onset and using a dose of ≤0.4 mg/kg/day significantly reduced relapse rates (1-year relapse 6.2%). This rate was similar to the group not receiving prednisolone. In contrast, early treatment initiation (0–6 weeks) was associated with the highest relapse rate (33.5%). In conclusion, although prednisolone generally increases the risk of relapse in GLM, delayed start and low-dose protocols appear to be the most appropriate treatment strategy to reduce this risk.

İntro:

The introduction provides a well-positioned, logical scientific basis. Clarify the purpose in the final section, aligning it with the study design.

MM&Results

Major Comments

The manuscript combines retrospective data (2017–2022) with prospectively collected data (from 2022 onward). This hybrid design introduces potential inconsistencies in data completeness and diagnostic/treatment protocols across time periods. The authors should clarify:

*how data collection was standardized across the retrospective and prospective phases,

*whether the two periods differ significantly in baseline characteristics or treatment strategies,

*whether a sensitivity analysis excluding early retrospective years was performed.

Prednisone administration was not randomized and may reflect clinician judgment based on disease severity. This introduces substantial bias:

*severe cases may have been preferentially treated earlier or with higher doses,

*the higher recurrence rate in the prednisone group may therefore reflect baseline severity rather than treatment effect.

Different analyses include different patient counts (e.g., 335 for timing analysis, 341 for duration, 337 for dose). The mechanism of missing data (MCAR/MAR/MNAR) is not described, and there is no imputation strategy. Given that missingness may introduce bias, the authors should:

*report missing data percentages for each variable,

*describe the missing data mechanism,

*and consider multiple imputation if appropriate.

The manuscript uses X-tile to identify cutoffs for treatment timing, dose, and duration. This method increases the risk of overfitting and inflated type I error, especially without an external validation cohort. The authors should clarify:

*whether correction for multiple testing was applied,

*whether cutoffs were validated using bootstrap or internal cross-validation,

*and how robust these thresholds are across sensitivity analyses.

Patients received diverse additional therapies, including antimicrobials, anti-tuberculosis agents, drainage procedures, and bromocriptine. Many of these are likely to influence recurrence risk. The multivariable Cox model includes some of these factors, but:

*their distribution between exposed and unexposed groups is not reported,

*no stratified or interaction analyses are provided,

*the possibility of treatment clustering (e.g., prednisone + drainage) is not explored.

*A more granular adjustment or subgroup analysis is warranted.

“Return of symptoms after clinical improvement or cure” is somewhat subjective and operator-dependent. Since recurrence is the primary endpoint, the authors should:

*specify objective criteria (e.g., imaging, physical examination, laboratory data),

*clarify whether recurrence was assessed by blinded observers,

*and report inter-observer variability if applicable.

Some subgroups used in the final survival analyses appear very small due to filtering (e.g., n=1 or n=2 cases added). This reduces reliability of hazard estimates. The authors should:

*report subgroup sample sizes clearly,

*refrain from overly granular stratification when numbers are insufficient,

*discuss the limitations in the power of subgroup comparisons.

Minor Comments

The dosing description (“initiated at a high dose and tapered by 5 mg every 3–7 days”) lacks:

*the definition of “high dose,”

*typical taper duration,

*criteria for tapering speed.

*A standardized protocol description is needed for reproducibility.

Figure 1 outlines the workflow but does not provide a CONSORT-style patient flow diagram. A flowchart of inclusions/exclusions and analysis subsets would greatly improve transparency.

The manuscript lacks a comparison of baseline characteristics between:

*prednisone vs. non-prednisone groups,

*short- vs. long-interval groups,

*low- vs. high-dose groups.

Rationale for Dividing Dosage by Both Absolute and Weight-Adjusted Methods: Both approaches are analyzed, but the rationale and clinical relevance should be explained more succinctly.

Language: The Methods section is thorough but dense. Some subsections may benefit from condensation for readability.

In summary;

This manuscript is strengthened by a large sample size and a comprehensive multidimensional analysis of prednisone therapy. However, concerns regarding confounding, missing data, exposure classification, and multiple threshold testing must be addressed before the conclusions can be fully supported. Clarification of the methodology and additional analyses as suggested above would considerably improve the manuscript’s rigor and impact.

Discussion:

The discussion is generally aligned with the study's three main objectives: treatment timing, dose, and duration; this is good.

However, although the reasons for choosing these three dimensions—the clinical relevance of the research gap—were strongly emphasized in the introduction, the discussion occasionally becomes disjointed. Some lengthy explanations (e.g., pathophysiology, steroid rebound mechanism) distract from the study's main question.

The comparison with the literature is comprehensive; many studies are compared. However, the comparisons are presented in very long passages, somewhat obscuring the clinical implications. In some sections, citations are listed consecutively, but the question "What does this study add to the literature?" is not clearly stated.

The proposed treatment strategy at the end of the discussion is a strong point; however, more targeted conclusions, such as for whom it should be applied and for which clinical phenotypes, would have added clinical value.

Limitations are accurately listed.

However, the heterogeneity of surgical interventions and the impact of nonsteroidal therapies, particularly as a critical limitation, should have been integrated into the discussion earlier.

Reviewer #3: Thank you for the opportunity to review this manuscript. IGM is a disease with many unknown issues, especially regarding its treatment. Studies including a large number of patients, like this one, could be very helpful in clarifying the concern of using steroids and their adverse effects in IGM.

The three dimensions considered for grouping treatment features and the cut-off point measurements are interesting, providing a well-designed analysis protocol. However, there are many concerns that need to be addressed before publication could be considered for this manuscript:

1. ABSTRACT: needs some English language editing. Also, the main concerns that will be approached below should be mentioned in brief in the abstract.

INTRODUCTION:

2. All through the introduction, the past tense is wrongly used for present data on GLM.

3. Lines 44 to 47: The authors mention that IGM incidence has rapidly increased during the present decade. For this, they cite a paper from 2019, which is a clinical study on the surgical treatment of GLM. That papers cite, for stating the incidence conditions, a paper from 2009, which reports a series of 54 cases of GLM. None of these investigates the incidence of GLM, and the year of publication of the latter is too old to talk about the recent decade. Therefore, this sentence is not supported by the present evidence. To update the issue, please know that the only papers published up to now (as far as I know) with an acceptable assessment of GLM incidence are those of the US CDC [Goldman M, et al. Idiopathic Granulomatous Mastitis in Hispanic Women--Indiana, 2006-2008. MMWR: Morbidity & Mortality Weekly Report. 2009 Dec 4;58(47)], which only includes Indiana (the city with the highest Hispanic population in the US) and cannot be applicated globally, and that of Zou et al [Zou J, et al. Clinical trends in granulomatous mastitis incidence, prevalence, and treatment: A retrospective study highlighting ethnic differences in care. Journal of Surgical Research. 2024 Oct 1;302:732-8.] which consists of a population-based study in the USA. The results of the latter are very valuable, but can only be applied to the US population, and they do not assess the recent incidence trends. So, the data in these lines of the manuscript lack evidence, unless the authors provide a correct reference.

4. Lines 55-57: ‘Although topical steroid therapy was reported as a useful treatment modality [6,11], its invasive nature limited its widespread clinical adoption.’ Topical steroids are the least invasive treatment, how do authors mention that the reason for its limited use is its invasive nature?

5. Lines 52-67: While using many references about it, the authors forgot to mention the recently adopted local (intralesional) use of steroids. May be, by topical use and its invasiveness, they meant local injections??

6. Introduction overall: The introduction is not well-written. Authors could have a better approach to the present therapeutic approaches and the existing challenges in GLM treatment. Also, despite the large amount of literature available, some references are selected inappropriately, like the use of references 1 (comparing local and oral steroids in GLM treatment) and 2 (review of GLM pathology) for the first sentence (‘ Granulomatous lobular mastitis was a rare, benign, and chronic inflammatory disease of the breast with an unknown etiology’).

METHODS

7. Lines 80-83: Before mentioning the number of excluded patients, the inclusion and exclusion criteria should be stated clearly.

8. Line 84: 1) What was the mode of diagnosis of GLM? What were the diagnostic criteria?2) Did the included patients used only prednisolone as their treatment modality? How did the authors get sure about the concomitant use of NSAIDs, or any combination treatment?

9. Please explain when were the lab tests done? As the test results were related to disease prognosis, the timing of the tests is very important. Usually, all GLM patients do not attend at the first disease presentation. Therefore, tests are taken at different disease stages. How was the condition of lab test timing in your database? If the mentioned concerns existed in your study (as expected), please mention this limitation in the discussion.

10. Fig.1: Although well-organized and very useful, I suggest that sentences are shortened. Usually, short sentences or expressions are used in these types of workflows. E.g., instead of ‘614 pts with GLM were ultimately included in the analysis’, a shortened version as ’Finally included: 614 patients’ would be more suitable for the figure. This should be done in other parts of the figure also. Importantly, the ’It was hypothesized that …’ is not appropriate for the figure. It could be replaced by something like ’Assessment of prednisolone effects’.

11. Line 144: How do authors explain the effect of the affected side on prognosis?

12. Tab 1: For prolactin, please provide the normal range. Also, define hyperlipidemia (which types of lipids? The ranges?). Was bromocriptine used for hyperprolactinemia? These should be described shortly in the footnote to make the table understandable and useful.

13. Tab. 1: For the age at diagnosis, which carried a worse prognosis, younger or elder age?

14. Lines 152-154: These descriptions should come in the Methods, not the results. Also, please explain what were the non-prednisone treatments. The overall treatment strategies are described in lines 97-100; however, we need to know, for example, if you had patients who only received antibiotics? Or only anti-tuberculosis agents? Or if bromocriptine was used alone, and regardless of the prolactin status? Regarding the higher recurrence in the prednisone group (as mentioned in lines 154-156 and Fig 2), it is crucial to know about the comparison group details.

DISCUSSION

15. Lines 283-285: The point is that the two groups: prednisone and non=prednisone are not compared regarding disease severity, so the results might just mirror the disease severity. Please explain if that is not the point, and otherwisw discuss the point as such and add it in the limitations.

16. Lines 309-327: These are hypothetical reasons for the higher recurrence in patients with early-onset treatment, and they do not follow a very sound rational. Patients who attended later probably had a less severe disease, and those that attended earlier had a more severe disease. So, treatment was begun sooner, but recurrence was higher. Although this is not certain, but it should be mentioned as a probable reason for these results, and recognized as a main confounding factor in the discussion and limitations.

17. Lines 328-331: Treatment duration depends on the disease response to treatment, so medicines cannot be cut before obtaining an acceptable response. Do the authors mean to discontinue steroids while the disease is improving but we are still waiting for recovery? The reason for the lower recurrence with a shorter treatment duration might simply be because of less severe disease. Authors should consider this important point in the discussion and limitations.

18. Lines 332 to 358: This part should be shortened and simplified with an accurate wording. Authors should pay attention to the tense of the verbs and the use of ‘this’ and ‘that’ for theirs vs. other studies.

19. Lines 367-369: Lower doses are most probably due to less severe disease, which explains the lower recurrence. As the clinical signs are not compared between the two groups, we cannot know whether the low and high dose groups were similar in this regard, and they were most probably not. This also applies to the weight-adjusted dose. This key point should be considered and alluded to in the discussion and limitations.

20. Apparently, the results and discussion indicate that the non-prednisone treatments, the lower doses of prednisone, the later-onset treatment initiation, and the shorter duration of prednisolone use is either better that or not different with the comparison group better. Wouldn’t all these reflect that prednisolone is better not used?

21. Overall, as disease severity has not been considered in the comparison between groups, results should be taken cautiously. The very positive sentences that present results of this study as doubtless data need to be re-written to reflect the uncertainty of the mentioned points. The limitations also should be revised accordingly.

**Do you want your identity to be public for this peer review?** For information about this choice, including consent withdrawal, please see our For information about this choice, including consent withdrawal, please see our Privacy Policy .

Reviewer #1: No

Reviewer #2: **Yes:** Irmak Durur-SubasiIrmak Durur-Subasi

Reviewer #3: **Yes:** Sadaf AlipourSadaf Alipour

---

## [Author Response · Author response to Decision Letter 1]

18 Dec 2025

For details, refer to Document Response to Reviewers

---

## [Editor Report · Decision Letter 1]

14 Jan 2026

The paradoxical effect of oral prednisone in granulomatous lobular mastitis: an observational cohort study.

PONE-D-25-57872R1

Dear Dr. Luo,

We’re pleased to inform you that your manuscript has been judged scientifically suitable for publication and will be formally accepted for publication once it meets all outstanding technical requirements.

Kind regards,

Siddharth Gosavi, MBBS, MD Internal Medicine,DNB Internal Medicine

Academic Editor

PLOS One

Additional Editor Comments (optional):

Satisfied with the response.
---

## [Editor Report · Acceptance letter]

PONE-D-25-57872R1

PLOS One

Dear Dr. Luo,

I'm pleased to inform you that your manuscript has been deemed suitable for publication in PLOS One. Congratulations! Your manuscript is now being handed over to our production team.

Kind regards,

on behalf of

Dr. Siddharth Gosavi

Academic Editor

PLOS One